# Emerging Role of the Mast Cell–Microbiota Crosstalk in Cutaneous Homeostasis and Immunity

**DOI:** 10.3390/cells12222624

**Published:** 2023-11-14

**Authors:** Cameron Jackson Bosveld, Colin Guth, Nathachit Limjunyawong, Priyanka Pundir

**Affiliations:** 1Department of Molecular and Cellular Biology, College of Biological Science, University of Guelph, Guelph, ON N1G 2W1, Canada; cbosveld@uoguelph.ca (C.J.B.); cguth@uoguelph.ca (C.G.); 2Center of Research Excellence in Allergy and Immunology, Research Department, Faculty of Medicine Siriraj Hospital, Mahidol University, Bangkok 10700, Thailand

**Keywords:** mast cells, microbiome, skin immunity, homeostasis, host–microbe

## Abstract

The skin presents a multifaceted microbiome, a balanced coexistence of bacteria, fungi, and viruses. These resident microorganisms are fundamental in upholding skin health by both countering detrimental pathogens and working in tandem with the skin’s immunity. Disruptions in this balance, known as dysbiosis, can lead to disorders like psoriasis and atopic dermatitis. Central to the skin’s defense system are mast cells. These are strategically positioned within the skin layers, primed for rapid response to any potential foreign threats. Recent investigations have started to unravel the complex interplay between these mast cells and the diverse entities within the skin’s microbiome. This relationship, especially during times of both balance and imbalance, is proving to be more integral to skin health than previously recognized. In this review, we illuminate the latest findings on the ties between mast cells and commensal skin microorganisms, shedding light on their combined effects on skin health and maladies.

## 1. Introduction

The skin is the largest organ in the body, with multifaceted duties. It protects the body from many types of damage, including ultraviolet radiation, temperature, microorganisms, toxins, and allergens [1,2]. There are many immune and resident cells associated with the skin, including keratinocytes, fibroblasts, Langerhans cells (LCs), dendritic cells (DCs), macrophages, αβ and γδT cells, natural killer T (NKT) cells, and mast cells. These cells all play different roles in the first line of defense, as well as in skin homeostasis [3,4]. The skin surface is also home to many commensal bacteria, fungi, and viruses, which together form the skin microbiome [5]. The skin microbiome plays a critical role in protection against pathogens, and if this homeostasis is disturbed, disease can follow [6]. The complexity of cutaneous immunity and microbiota interactions has symbiotic implications, but it may also continue to cause problems in today’s Western lifestyle, as autoimmune and inflammatory diseases are on the rise due to the effects of the skin microbiota on the immune cells residing there [5]. Among the immune cells in the skin, mast cells are tissue-resident innate immune cells filled with secretory granules designed to initiate a pro-inflammatory response and recruit other innate and adaptive immune cells [4,7]. This review will specifically focus on the interactions between mast cells and the cutaneous commensal bacteria, shedding light on the intricate relationship between the immune system and the microbiome in the context of the skin.

## 2. Skin Innate Immunity

Skin innate immune cells play a vital role in the body’s defense against pathogens and in maintaining skin health. Within the skin, immune cells are mostly located in the dermis and provide the first line of defense against invading microorganisms. A complex organization and a diverse immune cell population are involved in skin immunity and have been reviewed elsewhere for detailed discussions [8,9,10]. Herein, we focus only on mast cells, specialized innate immune cells at the forefront of immunity and inflammation. The skin contains the highest percentage of mast cells, with mast cells making up to 10% of the leukocyte population in the ear skin of the mouse [7,11]. Mast cells hold particular significance in skin immunity for two main reasons: (a) Mast cells possess a spatial advantage by strategically positioning themselves near blood vessels and nerves, enabling them to quickly detect and respond to foreign substances, above all facilitating communication with blood vessels’ cells such as endothelial cells, pericytes, and sensitive neurons. (b) Additionally, their ability to store and release densely packed secretory granules within seconds of activation grants them a temporal advantage, allowing for a rapid and amplified immune response when needed [12]. Overall, mast cells have a strong connection with the epithelium, playing a significant role in maintaining epidermal barrier function and skin homeostasis [13,14]. Numerous studies emphasize the significance of intercellular communication between mast cells and nearby immune and nonimmune cells of the skin in upholding barrier function and immune balance [4]. Consequently, it is crucial to closely regulate the body’s reactions to commensal bacteria, another resident of the skin compartment, in order to avoid the development of pathology resulting from unnecessary immune activation. As such, mast cells help maintain a delicate balance by recognizing and tolerating skin commensal bacteria while remaining vigilant against potential pathogens. Mast cells can modulate immune responses and promote immune tolerance, ensuring a harmonious coexistence with the commensal bacterial community. We will further discuss the intricate connection between commensal bacteria and skin-resident mast cells, unraveling its impact on mast cell development, function, and the balancing act of health and disease.

## 3. Mast Cells in the Skin—A Brief Overview

### 3.1. Mast Cell Subtypes and Development

Although all mast cells are remarkably distinct from other leukocytes based on their unique staining with cationic metachromatic dyes as first identified by Paul Ehrlich, they exhibit multiple phenotypic variations that stem from different origins and tissue microenvironments of their respective tissue locations, resulting in different functional specializations. These subsets display distinct cytokine expressions, granule contents, and receptor profiles (Table 1). Murine and human mast cells also differ in their expression of receptors and their naming scheme. In mice, there are two main mast cell subclasses defined based on their anatomic location at maturity: connective tissue mast cells (CTMCs) found in most connective tissues (e.g., the skin, peritoneal cavity, trachea, tongue, esophagus, etc.) and mucosal mast cells (MMCs), primarily situated inside the respiratory and intestinal mucosa. In humans, these are equivalent to tryptase^+^ chymase^+^ mast cells (MCTCs) and tryptase^+^ mast cells (MCTs). As the name suggests, the most obvious difference between mast cell phenotypes is the production of both tryptase and chymase or the production of solely tryptase within their releasable granules. Recently, Tauber et al. used unbiasedly single-cell RNA sequencing (scRNA-seq) to analyze molecular differences in both murine and human mast cell populations across multiple organs and found that the heterogeneity of human mast cells was far more complex than that of murine mast cells [15]. Based on transcriptomic profiles, they illustrated seven potential mast cell subsets distributed in different organs of humans, of which three different subsets were specifically found only in the skin.

Furthermore, it is evident that CTMCs and MMCs may have discrepancies in their developmental origins (Figure 1). While the majority of CTMCs arise from yolk-sac-derived erythro-myeloid progenitors (EMPs) that migrate and develop in the fetal liver during embryogenesis and are constitutively present, long-lasting, and self-maintain independently from bone marrow precursors in most connective tissues throughout life, MMCs are short-lived, with a lifespan of only 2 weeks, and can be readily replaceable with bone-marrow-derived agranular hematopoietic stem cell (HSC) progenitors [7,16,17,18,19]. Of note, several bone marrow adoptive transfer studies have affirmed that relative to MMCs, CTMCs in the skin of adult mice exhibit much poorer reconstitution of donor-derived HSCs, suggesting the self-renewal of long-lived, tissue-resident precursors of mast cells in the skin [18,20]. Regardless of its origins, mast cell development is influenced by tissue-specific growth factors such as stem cell factor (SCF), transforming growth factor-β, CCL2, IL-3, activin, and more [21,22].

### 3.2. Anatomic Location of Mast Cells in the Skin

Most mast cells in the skin are located in the superficial dermal layer, below the outer layer of the skin, and close to nerves, blood vessels, hair follicles, adipose tissue, and muscle tissue [3,21]. The anatomical relationship between mast cells and nerve fibers is observed in various organs. In the skin, mast cells are found in close proximity to C- and A-type peripheral nerve fibers that express receptors for the mast cell mediator, histamine [23]. Conversely, mast cells express a multitude of receptors for classical neurotransmitters (e.g., acetylcholine and corticotropin-releasing hormone) and neuropeptides (e.g., substance P and calcitonin gene-related peptide). This interaction between mast cells and nerves forms a feedback loop, where histamine released from mast cells triggers the release of neuropeptides. These neuropeptides, in turn, stimulate mast cell degranulation, leading to the release of histamine and prostaglandin D_2_ (PGD_2_), thus perpetuating the cycle [23]. 

Similarly, the anatomical association and resulting interactions between blood vessels and skin mast cells are multifaceted. These interactions encompass the connection with blood endothelial cells (BECs) as well as the interaction with cells in the circulation [4,21]. Histamine release from skin mast cells leads to increased blood flow, changes in vascular endothelial cadherin (CD144) localization, and the hyperpermeability of the vasculature [4,24]. Mast cells release TNF⍺ into the bloodstream through the vessel wall to recruit additional neutrophils, specifically during firm adhesion and intraluminal crawling [25]. Additionally, mast cells produce multiple vascular endothelial growth factors (VEGF-A, -B, -C, and -D), contributing to both angiogenesis (VEGF-A and -B) and lymphangiogenesis (VEGF-C and -D) [4,25]. Another way mast cells interact with blood vessels is by extending filopodia through the blood vessel wall. Perivascular mast cells can capture IgE, serving as a sampling mechanism and possibly recruiting other cells as they circulate through the system [26,27]. Skin mast cells also interact with muscles by inducing contraction through leukotrienes, and they secrete interleukins as immune cell recruiters and activators, triggering an inflammatory response [21]. 

Mast cells in the dermis are also found in direct or indirect contact with other cutaneous cells, such as dendritic cells (dermal DCs and Langerhans cells), dermal macrophages, innate lymphoid cells (ILC)2, fibroblasts, keratinocytes, and melanocytes [28]. Several of these structural and immune cellular components of the skin, including keratinocytes, endothelial cells, smooth muscle cells, and fibroblasts, as well as eosinophils and other mature mast cells, are found to be the source of SCF, one of the most important factors for mast cell maturation, proliferation, the inhibition of mast cell apoptosis, inducing chemotaxis, adhesion, and increased degranulation [29,30]. Additionally, contact with fibroblasts has been observed to contribute to an increased expression of genes associated with the connective tissue mast cell phenotype. In turn, mast cells can secrete IL-4, IL-13, and fibroblast growth factors to promote fibroblast proliferation [31,32] or release histamine to initiate collagen production from fibroblasts [33]. Cutaneous mast cells are also found to directly interact with dermal DCs in both contact-dependent and contact-independent manners (reviewed in ref. [28]). Furthermore, there is compelling evidence indicating that mast cells play a significant role in promoting the growth of melanocytes, which are specialized pigment-producing cells in the skin, as well as melanoma cells. This promotion occurs through the action of mediators like fibroblast growth factor 2 (FGF-2) and IL-8, thereby contributing to the progression of tumors [34,35]. Additionally, histamine, released by mast cells, affects the melanogenesis, migration, and morphology of melanocytes via the H2 receptor. UV radiation can also trigger histamine release, potentially driving hyperpigmentation. These histamine effects on melanocytes and vitiliginous keratinocytes may support histamine’s use in repigmentation for vitiligo patients [36,37,38].

### 3.3. Phenotypic Characteristics of Skin Mast Cells

In general, mast cells are unique and can be easily characterized by their high content of electron-dense lysosome-like secretory granules within their cytoplasm. These secretory granules contain a plethora of preformed mediators, including various lysosomal enzymes, mast-cell-specific as well as non-mast-cell-specific proteases, biogenic amines such as histamine and serotonin, cytokines, and growth factors [39]. Additionally, mast cells can be readily identified using flow cytometry by their co-expression of cell surface markers: the high-affinity receptor for immunoglobulin E (IgE), FcεRI, and SCF receptor, c-kit (CD117). Likewise, their distinct transcriptomic profiles set them apart from other immune cells as well [7]. However, as mentioned earlier, mast cells have multiple subsets, each showing variations in phenotypes and functions, including the production of cytokines, chemokines, and complement receptors, as summarized by Elieh et al., Xing et al., and McNeil et al. relating to mice and humans [40,41,42]. When focusing on skin mast cells, they share many transcriptional signatures with other connective tissue mast cells, particularly those found in the peritoneum [7]. It has become evident that these CTMCs, including cutaneous mast cells, display significant enrichment in genes responsible for serine proteases (e.g., *Ctsg* (encoding for cathepsin G), *Mcpt2* (for mast cell protease 2), *Mcpt4*, *Mcpt9*, *Tpsab1* (for tryptase alpha/beta 1), *Tpsb2*, *Tpsg1* (for tryptase gamma 1), *Cma1* (chymase 1), and *Cma2*) as well as prostaglandin synthases, which are pivotal enzymes in eicosanoid/prostaglandin biosynthesis (e.g., *Hpgds* (encoding for hematopoietic prostaglandin D2 synthase)) [7]. Another key molecular signature of these CTMCs is their exclusive expression of a member of Mas-related G-protein-coupled receptors (MRGPRs): human *MRGPRX2* and its murine orthologue *Mrgprb2* (Figure 1) [7,43]. MRGPRX2/Mrgprb2 have garnered increased attention over the past decade due to their responsibility for IgE/FcεRI-independent degranulation of mast cells in response to a wide repertoire of basic secretagogue molecules or cationic peptides such as neuropeptides, antimicrobial host defense peptides (HDPs), compound 48/80 (c48/80), and several FDA-approved drugs, which can induce potentially life-threatening pseudoallergic reactions [41,44,45,46,47,48]. Recent scRNA-seq analysis has verified that the expression of *Mrgprb2* is restricted to CTMCs, including skin mast cells, and MrgprB2^+^ vs. MrgprB2^−^ mast cells represent distinct mast cell subsets with a conserved transcriptomic core consistently across different tissues in mice [15].

## 4. Cutaneous Microbial Diversity—Healthy vs. Inflamed Skin

### 4.1. The Cutaneous Microbiome in Health

The cutaneous microbiota is composed of a diverse range of microbes, encompassing bacteria, fungi, and viruses, referred to as commensal organisms. Undoubtedly, bacteria are the most prevalent microorganisms distributed across various skin sites. 16*S* ribosomal RNA (rRNA) sequencing has revealed that microbiome bacterial residents typically fall into three main genera: *Corynebacteria, Propionibacteria,* and *Staphylococci* [49]. While these are generally the most common genera, population frequencies tend to be more variable when comparing skin sites with different environments that create specialized niches. For example, sebaceous regions, such as the side of the nose and back of the scalp, where large amounts of sebum are produced, tend to have higher amounts of the lipophilic *Propionibacterium,* whereas moist areas like behind the knee have more *Corynebacteria* and *Staphylococci* which are equipped to use the abundant amounts of sweat as a resource [50]. On the contrary, the constitution of the fungal community, determined through sequencing of the internal transcribed spacer 1 (ITS1) region of the eukaryotic ribosomal gene, remains relatively consistent along distinct topographical skin sites. *Malassezia* spp. fungi are the predominant species in most regions, irrespective of their physiological attributes, although a greater diversity of fungi community could be exhibited in some areas such as foot sites as well [51,52]. Unlike bacteria and fungi, the investigation of viral community diversity presents unique challenges principally due to the absence of a universal marker gene shared among these microorganisms, and the presence of a eukaryotic virus has been found to be individual-specific rather than being tied to specific anatomical sites [51].

When healthy, the skin microbiome is largely stable; however, changes in the pH, moisture, and physiologic composition of the skin influence the bacteria that reside there [51,53]. These differences include, for example, dry, moist, or sebaceous microenvironments [5,51]. Diversity and community stability are inversely correlated, as diversity increases, stability decreases. In healthy individuals, the skin microbiota is generally homeostatic, and if the community changes over time, species may be replaced with other microbes of similar niches and in similar numbers, resulting in overall little change in the population as a whole. Individuals will have differences in their skin microbiomes, especially in moist sites, whereas sebaceous and dry sites are more similar between individuals [51]. 

While the commensal organisms of the microbiota confer protection against pathogens and disease, some organisms have the potential to cause harm to the host if left unchecked. In order to prevent this, certain organisms can regulate the growth of others to maintain homeostasis. For example, *Staphylococcus aureus* is an opportunistic pathogen capable of forming biofilms and causing atopic dermatitis, also known as eczema, if given the opportunity [54]. To prevent this from happening, the commensal organism *Staphylococcus epidermidis* secretes a serine protease known as Esp which can degrade proteins used by *S. aureus* to form biofilms, which, in combination with antimicrobial peptides secreted by other skin residents, prevents the harmful effects of *S. aureus* [54]. 

In addition to regulating the growth of pathogens via their own means, commensal organisms can synergize with resident immune cells to confer immunity. In a study comparing the release of inflammatory cytokines known to initiate and strengthen the immune response, interferon-gamma (IFN-γ) and interleukin (IL)-17A, in specific-pathogen-free (SPF) and germ-free (GF) mice, it was found that T cells of GF mice produced significantly less IFN-γ and IL-17A compared with SPF mice, resulting in impaired skin immunity essential for controlling infection by other pathogens in GF mice [55]. Notably, this reduction in T cell cytokine production capacity was primarily attributed to the lack of skin microflora, as the presence of other immune cells in the skin and skin-draining lymph nodes remained comparable between GF and SPF mice, and the depletion of microflora, specifically in the intestine, but not in the skin, via oral antibiotic treatment had no impact on the production of these inflammatory cytokines by cutaneous T cells. Interestingly, the introduction of only a single skin commensal bacteria, such as *S. epidermidis*, proved effective in restoring IL-17A production in the skin [55]. Overall, these findings indicate that resident commensal microorganisms are vital for maintaining immune fitness within the skin. Another example exhibiting this synergy between the microbiome and immune response is found in wound healing. During the wound healing process, a lipoteichoic acid produced by *S. epidermidis* can prevent the excessive release of pro-inflammatory signals from keratinocytes, thus promoting a controlled wound healing process and preventing the overactivation of the immune system [56]. 

### 4.2. The Cutaneous Microbiome in Disease

Many common skin diseases such as atopic dermatitis, allergic contact dermatitis, acne, chronic wounds, and chronic inflammation via the immune system’s activation have been shown to be associated with changes in the skin microbiome, known as dysbiosis or dysbacteriosis [6]. Healthy individuals can lose control of their skin microbiome from environmental exposure, host factors, and/or bactericidal product secretion, resulting in a decrease in the diversity and stability of the cutaneous microbiome [51,53]. For instance, in atopic dermatitis, *S. aureus* populations increase above normal, which is associated with disease flares and worsening symptoms [57]. When these population shifts occur, the immune system is equipped to recognize the threat and respond. One mechanism in the response of the immune system in atopic dermatitis is the degranulation of mast cells in response to δ-toxin produced by *S. aureus*, inducing both innate and adaptive immune responses [24]. While the role of dysbiosis has been studied much more extensively in atopic dermatitis than in other diseases, evidence suggests that there may be a connection between dysbiosis and other diseases, such as psoriasis. Individuals with psoriasis have differences in skin microbiome populations compared with healthy individuals, but this relationship was not determined to be conclusive [58]. Further investigation is required to validate this association, as well as associations between dysbiosis and other cutaneous diseases and infections. 

Together, the skin microbiota and immune networks form a sophisticated defense system that protects against pathogens, maintains skin integrity, and regulates immune responses in the skin. Understanding the interplay between these components is essential for developing effective strategies to promote skin health or treat skin-related disorders. 

### 4.3. Mast Cells Interact with Commensal Bacteria

The conventional, well-described pathways of mast cell activation are illustrated in Figure 2. While it is established that mast cells reside in the dermis [3,21], their direct interaction with bacterial pathogen-associated molecular patterns (PAMPs) [2,3,59,60] suggests there must be a mechanism via which bacteria can penetrate into the deeper layers of the skin. Nakatsiki et al. and Grice et al. showed that bacteria do extend into the dermal layers of the skin as they detected commensal bacteria in subcutaneous regions of normal healthy human skin with no site of injury [49,61]. Although the paper only studied the presence of DNA encoding for 16S rRNA genes, specific antigens, and bacterial rRNA, they could not prove live bacteria were present, and their components show that mast cells and other immune cells located in the dermis have access to direct activation by bacterial PAMPs. A study by Bay et al. also showed that skin bacteria have the ability to infiltrate healthy skin, although at a lower operational taxonomic unit (OTU) species richness level [62]. This shows that although the species richness decreases in the dermal layers of the skin, there is still the possibility for mast cells and other immune cells to interact directly with bacteria in the skin [62].

## 5. The Role of the Skin Microbiome in Mast Cell Development

The role of the skin microbiome in mast cell maturation is of the utmost importance. This relationship has been shown most pertinently in a study by Wang et al. comparing mast cell maturation in GF and SPF mice. In that study, mast cell maturity was defined as high expression of the SCF receptor, c-kit, while immaturity was defined as low c-kit expression. GF mice were found to have significantly fewer mature mast cells in addition to reduced concentrations of SCF in the skin compared with their SPF counterparts [63]. Additionally, this reduction in mature mast cells was recovered when GF mice had their microbiota reconstituted via exposure to SPF mice. Further emphasizing this lack of maturity, it was found that hind paw inflammation, induced by injection with the common mast cell activator compound 48/80, was reduced in GF mice compared with SPF mice [63]. To elucidate the mechanism behind this discovery, Wang et al. showed that injection with staphylococcal lipoteichoic acid has the ability to upregulate SCF production in keratinocytes, which was, in turn, able to increase the number of c-kit-expressing mast cells in both GF and SPF mice. Furthermore, the keratinocyte-specific knockout of the *Scf* gene using a Cre-lox system completely abolished mast cell recruitment to the skin of these mutant mice, clearly displaying that mast cell migration is entirely dependent on keratinocyte SCF production [63]. The skin microbiome plays a pivotal role in controlling mast cell homing and their maturation in the skin. This regulation occurs through the modulation of keratinocyte-derived SCF production in response to staphylococcal lipoteichoic acid (Figure 3). This was initially demonstrated by the observation that overexpressing SCF in keratinocytes leads to an increase in the mast cell population in the skin [63]. That study was preceded by the studies by Kunisada et al. and Huttunen et al., who set out to identify the roles of keratinocytes, mast cells, and SCF in wound healing, albeit without the elucidation of the mechanisms [64,65]. While these studies provide useful insights into the mechanisms behind the relationship between the skin microbiome and mast cell maturation, there is still a lot to discover about this relationship, such as what other bacteriologically derived molecules can influence this system. 

## 6. Mast Cells in Skin Barrier Function—Germ-Free vs. Conventional Mice

Mast cells are responsible for epidermal barrier function and wound healing [66,67,68]. The microbiome plays a pivotal role in alerting the immune system of a breach in the skin barrier. As the skin is breached under normal circumstances, the microbiome releases PAMPs, which are then detected by resident immune cells, such as mast cells. When the microbiome is not present (such as in GF mice), the immune system initially can only be recruited to the site of injury by danger-associated molecular patterns (DAMPs) from injured cells in the epidermal layer of the skin [66]. Since the immune system is not primed by the microbiological foreign body and pathogen-associated mediators, the immune system is considered immature. This results in a slower initial response to wounds in the skin barrier [66]. Mast cells are required for normal wound healing in mice, which may be extrapolated to humans [68]. Mast cells release a plethora of VEGF growth factors, which are used for promoting angiogenesis. This release of VEGF also lasts longer in GF mice than in conventional SPF mice [66]. Mast cells also produce IL-10, which can create a feedback loop with other mast cells. This feedback causes the downregulation of FcεRI receptors, resulting in the suppression of inflammatory factor release via IgE activation [66]. 

## 7. Mast Cell Tolerance to Commensal Bacteria

Mast cells exhibit a close relationship with the epithelium, playing a role in supporting barrier function. Consequently, the reactions with commensal bacteria necessitate meticulous regulation to avert the potential consequences of unwarranted immune activation. Recently, Di Nardo et al. uncovered important links between mast cells and dermal fibroblasts (dFBs) [69]. They showed in vitro co-culturing of mast cells and dFBs, resulting in mast cells becoming tolerant to commensal bacteria. dFB-conditioned human mast cells downregulated the production of pro-inflammatory cytokines such as IL-6, IL-8, and Th2 cytokines IL-4, IL-5, IL-10, and IL-13. The interactions between mast cells, commensal bacteria, and dFBs depend on each other. Mast cells need the microbiome to mature, and the microbiome needs dFBs to regulate mast cells in the skin. The group then observed differences in a dFB—mast cell co-culture using scRNA-seq. They observed phenotype switching in dFBs and the downregulation of immune-activating genes in mast cells. NF-𝜅B inhibitors were also upregulated in mast cells, namely, A20/tumor necrosis factor α-induced protein 3 (TNFAIP3) and NF-κB inhibitor alpha (NFKBIA). RNA-seq and in silico analysis pointed to CD44 and connections with the extracellular matrix (ECM). Toll-like receptor-2 (TLR2) is the associated receptor for CD44 and was, therefore, a target of the next phase of the study. Hyaluronic acid (HA) is a proposed ligand of TLR2 in specific tissues and was a focus of TLR2 activation in human mast cells. TLR2 was found to be removed from the cell surface when in the presence of HA, but TLR4 and CD44 remained expressed. Co-culture with dFB decreased CD44 [69]. Moreover, the researchers identified a key signaling pathway, the TGF-β pathway, mediating communication between fibroblasts and mast cells. Activation of this pathway promotes mast cell tolerance and dampens inflammatory responses. The findings of the study further shed light on the intricate interactions between skin cells and mast cells in maintaining immune homeostasis [69]. To summarize, the study uncovers a pathway in which a component of the ECM, HA, inhibits mast cell responses to commensal bacteria and reduces reactions to skin pathogens by suppressing the NF-κB pathway via TLR2 downregulation (Figure 4). While the research provides insights from in vitro and ex vivo studies, confirmation in human in vivo experiments is lacking. The study primarily focuses on the interaction between mast cells and dFBs and suggests the need for further investigation into the influence of other cell types, like neurons, on mast cell activation.

Similarly, products derived from epithelial or endothelial cells could potentially coordinate mast cell tolerance toward skin microbial communities. Interleukin-33 is a nuclear cytokine abundantly expressed in epithelial cells, endothelial cells, and fibroblast-like cells, both in homeostatic and inflammatory conditions. It operates as “alarmin,” released upon cellular injury or tissue damage, to alert immune cells expressing the ST2 receptor [70]. Mast cells are one of the primary targets of IL-33, and the interaction between the two is pivotal to allergic, infectious, and chronic inflammatory diseases. Intriguingly, during homeostasis, IL-33 renders mast cells unresponsive to the bacterial cell wall components lipopolysaccharide (LPS) and peptidoglycan (PGN). The inhibitory effect on LPS- and PGN-induced mast cell activation is observed at suboptimal concentrations of IL-33 and mediated via the ST2 pathway. Mast cells derived from ST2-deficient animals are hyperactivated by LPS, suggesting that IL-33 inhibits mast cell activation during immuno-homeostasis in vivo. Mechanistically, minimal IL-33 concentrations prompt the degradation of interleukin-1 receptor-associated kinase 1 (IRAK1) in mast cells, incapacitating them from reacting to LPS, hence averting immune responses to commensal bacteria. Conversely, during instances of infection or tissue damage, elevated IL-33 levels activate mast cells, prompting the release of proinflammatory cytokines and chemokines [71]. 

Understanding the mechanisms behind mast cell tolerance to commensal bacteria can have implications for developing therapies for inflammatory skin disorders, such as atopic dermatitis and psoriasis, where mast cell activation plays a significant role. For example, a study by Yu et al. investigated the influence of skin commensal bacteria on skin structure and mast cell levels. Findings revealed that the skin tissue of the control group displayed a normal, undisturbed structure without dermal anomalies. Conversely, the atopic dermatitis group exhibited substantial skin impairment, characterized by a thickened epidermis, hyperkeratosis, acanthosis, swelling of epidermal cells, dilation of dermal blood vessels, and notable infiltration of inflammatory cells. Remarkably, when S. *epidermidis* was applied to skin lesions in the atopic dermatitis group, the mice showcased mitigated skin impairment compared with the control group. Evaluation via toluidine blue staining further demonstrated a significant rise in the mast cell count in the atopic dermatitis group relative to the normal group. Noteworthy was the decrease in the mast cell count within the atopic dermatitis + *S. epidermidis* group compared with the atopic dermatitis group. Overall, these outcomes suggest that skin commensal bacteria possess the capacity to ameliorate skin damage and alleviate the severity of atopic dermatitis in mice [60]. 

Several commensal bacteria have demonstrated the ability to suppress mast cell degranulation via both TLR-dependent and TLR-independent pathways [72,73,74]. The latter pathway involves the inhibition of intracellular signaling of FcεRI by *Escherichia coli* [75], *Lactobacillus* [76,77,78,79], and *Bifidobacterium* [80]. Diverse strains of *Lactobacillus* have exhibited the capacity to mitigate allergic dermatitis in mouse models [81,82,83,84,85,86,87]. More recently, these findings have been extrapolated to human clinical trials, investigating the clinical effectiveness of probiotics in pediatric and adult patients with atopic dermatitis. The regular consumption of a blend of probiotics (*Lactobacillus* and *Bifidobacterium*) over a span of six months demonstrated a significant reduction in SCORing Atopic Dermatitis (SCORAD) scores among children and adolescents [88]. The possible utilization of *Lactobacillus* in the management and prevention of atopic dermatitis was recently summarized by Xie et al. (2023) [89]. 

## 8. Commensal Bacteria Prime Mast Cells against Pathogens

The interaction between the skin microbiota and mast cells goes beyond mere development and maturation, extending to the enhancement of mast cell immune function against potential pathogens. A notable example is the role of lipoteichoic acid (LTA) derived from the skin commensal *S. epidermidis*, which orchestrates the recruitment of mast cells to the skin’s surface. *S. epidermidis*, the most prevalent Gram-positive bacterial species on the skin, expresses LTA, a ligand for TLR2. Activation of TLR2 signaling through LTA prompts mast cells to upregulate the expression of cathelicidin antimicrobial peptides. The consequences are noteworthy: mast cells pre-conditioned with LTA exhibit heightened resistance to vaccinia virus infection. This heightened antiviral activity can be primarily attributed to the cathelicidin produced by mast cells in response to LTA-induced TLR2 signaling. As a result, the presence of these commensal components on the skin surface endows mast cells with a constant state of readiness against potential invading pathogens. This not only bolsters their ability to counter infections but also primes them to act as vigilant sentinels at the body’s point of entry [90]. 

## 9. Intra- and Interspecies Communication of Bacteria Influences Mast-Cell-Mediated Cutaneous Inflammation

Bacteria employ various mechanisms to exchange information and coordinate their behavior with other species, highlighting their ability to form intricate communities. One prominent form of communication is quorum sensing, where bacteria release and detect specific chemical signals called quorum-sensing molecules (QSMs). QSMs are typically small molecules, such as acyl-homoserine lactones in Gram-negative bacteria and autoinducing peptides in Gram-positive bacteria. These molecules accumulate as bacterial populations grow, enabling cells to gauge their own density. Once a threshold concentration is reached, coordinated activities such as biofilm formation, virulence factor production, and gene expression are initiated (reviewed in [91,92,93]). Mast cells have been reported to interact with cationic QSMs produced by Gram-positive bacteria via the receptors Mrgprb2 (mouse orthologue) and MRGPRX2 (human orthologue) [41]. This resulted in the activation and subsequent degranulation of mast cells and bacterial clearance (Figure 5). In addition to QSMs such as competence-stimulating peptides (CSP-1/-2), enterobactin synthase component F (Entf), and streptin-1, MRGPRX2/b2 is also reported to interact with multiple antimicrobial peptides and host-defense peptides, as summarized by Corbiere et al. [45,94].

Intriguingly, research conducted by Williams et al. on the development and control of atopic dermatitis revealed that interspecies quorum sensing between symbiotic and pathogenic bacteria on human skin plays a defensive role by restraining the damage caused by *S. aureus*. During microbial dysbiosis, *S. aureus* impairs the epidermal barrier through the activity of phenol-soluble modulin (PSM)α peptides. Exposure to these peptides leads to increased enzymatic activity in the epidermis, causing the breakdown of the skin barrier and subsequent inflammation. However, when coexisting with the commensal microflora, particularly CoNS (coagulase-negative Staphylococci), the detrimental impact of *S. aureus* on the skin is mitigated. This is due to the presence of the quorum-sensing agr system. Type I autoinducing peptide, a component of the agr system, inhibits *S. aureus* agr activity, thereby alleviating skin inflammation induced by *S. aureus*. During atopic dermatitis flares, when dysbiosis is severe, the abundance of these inhibitory type I peptides becomes insufficient, allowing *S. aureus* to contribute to inflammation. This is supported by the presence of PSMα on atopic-dermatitis-affected skin. These findings uncover a novel mechanism through which various members of the skin microbiome can counteract the disease-promoting effects of *S. aureus*. This could potentially explain why an overabundance of *S. aureus* and a decrease in bacterial diversity is linked to more severe atopic dermatitis symptoms. Considering the pivotal role of mast cells in atopic dermatitis pathophysiology and the possibility of harnessing them to benefit the host by intercepting bacterial quorum-sensing communication, comprehending the interplay between mast cells and the skin microbiome in states of harmony and imbalance could pave the way for enhancing current therapeutic approaches.

## 10. Conclusions

In summary, this review provides an up-to-date overview of the current understanding of the interplay between mast cells and the bacterial microbiota in the skin, highlighting its influence on the regulation of the cutaneous immune system and host homeostasis. The role of mast cells in this context undoubtedly varies depending on their specific interactions with different microorganisms. The mechanisms by which certain components of the microbiome fine-tune mast cell tolerance and function are only beginning to be unraveled. This emerging field of host–microbe interactions presents numerous exciting opportunities for future research, with the potential to pave the way for the development of novel therapeutic approaches for combatting infectious and inflammatory diseases. 

Skin mast cells, among innate immune cells, exhibit remarkable longevity, with lifespans of up to 12 weeks, and possess the ability to replenish and modify their granules following activation. This unique feature grants mast cells an inherent short-term memory when encountering pathogens, danger signals, and potentially resident microflora. This memory concept, akin to the “trained immunity” observed in other innate immune cells, has the potential to influence mast cell responses during subsequent encounters, thereby impacting both protective and allergic reactions. The underlying mechanism responsible for this innate immune memory is thought to involve epigenetic reprogramming, encompassing processes such as histone modifications, DNA methylation, and the expression of specific microRNAs and non-coding RNAs. Collectively, these mechanisms reshape the cell’s transcriptional program upon stimulation (as reviewed in [95]). However, there is a paucity of research examining long-term changes in mast cell functional programs in response to stimuli.

For instance, mast cells can exhibit short-term memory after LPS stimulation, similar to endotoxin tolerance in macrophages. Some interactions between IgE and LPS stimulation in mast cells have also been observed. IgE-induced sensitization primes mast cells for higher response to LPS via the pre-activation of NF-κB transcription factor [96]. A more recent study revealed that while IgE and β-glucan stimulation did not induce tolerance or training in mast cells, LPS conditioning led to significant and enduring changes in signaling pathways. LPS resulted in a state of unresponsiveness to secondary LPS stimulation by impairing the PI3K-AKT signaling pathway, leading to reduced NF-κB activation and decreased TNF-α and IL-6 release compared with naïve mast cells. Additionally, LPS-primed mast cells exhibited heightened TNF-α release when exposed to live *Candida albicans*, suggesting LPS can induce both tolerance and training responses depending on the subsequent challenge. Notably, inhibiting HDAC during LPS stimulation partially restored the response of LPS-primed mast cells to a secondary LPS challenge but did not reverse their increased cytokine production when exposed to *C. albicans* [97]. This demonstrates that mast cells, like other innate immune cells, can develop innate immune memory, and different stimulatory conditions can influence whether mast cells dampen or enhance the local inflammatory response. Still, whether trained immunity applies to mast cells for various stimuli, its in vivo relevance, and the underlying mechanisms remain unclear. Further investigation is needed to understand the potential role of trained immunity in modulating mast cell responses, especially in the context of the crosstalk with the commensal bacteria. 

## Figures and Tables

**Figure 1 cells-12-02624-f001:**
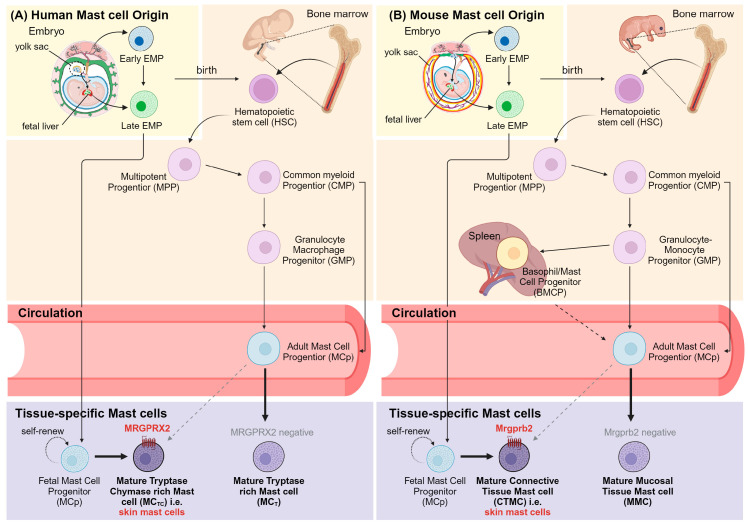
Mast cell development in humans and mice. (**A**) Human tissue-specific mast cells either develop during embryogenesis in utero, migrate to the target sites, and self-maintain via progenitors in the tissue, or develop postnatally in bone marrow and renew via committed mast cell progenitors through circulation. (**B**) Mouse mast cell development starts the same way in the embryo or bone marrow. Several lines of evidence suggest bipotential basophil–mast cell progenitors (BMCPs) are capable of differentiating into either basophils or mast cells in murine spleen as well. Figure created with BioRender.

**Figure 2 cells-12-02624-f002:**
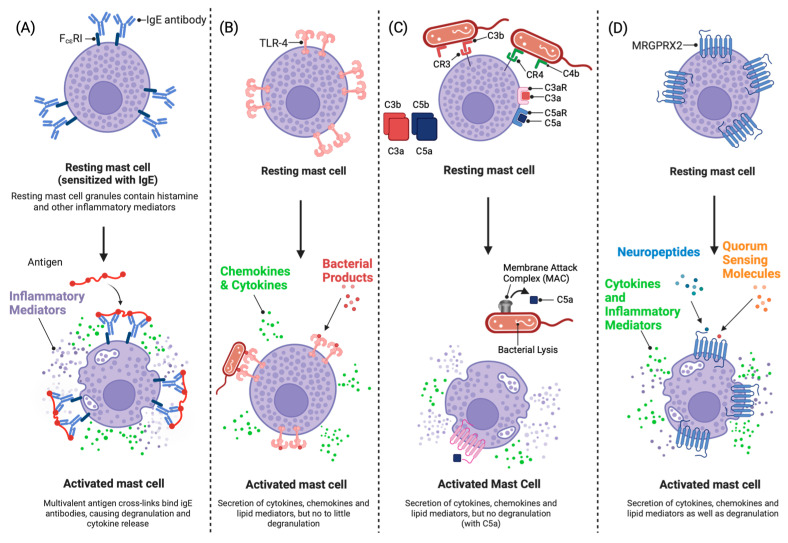
Mast cell activation by different stimuli. (**A**) IgE-mediated mast cell activation via FcεR1. (**B**) Non-IgE-mediated mast cell activation via Toll-like receptors (TLRs) in response to PAMPs and DAMPs. (**C**) Non-IgE-mediated mast cell activation via complement. (**D**) Activation by neuropeptides and quorum-sensing molecules via the MRGPRX2 receptor. Figure created with BioRender.

**Figure 3 cells-12-02624-f003:**
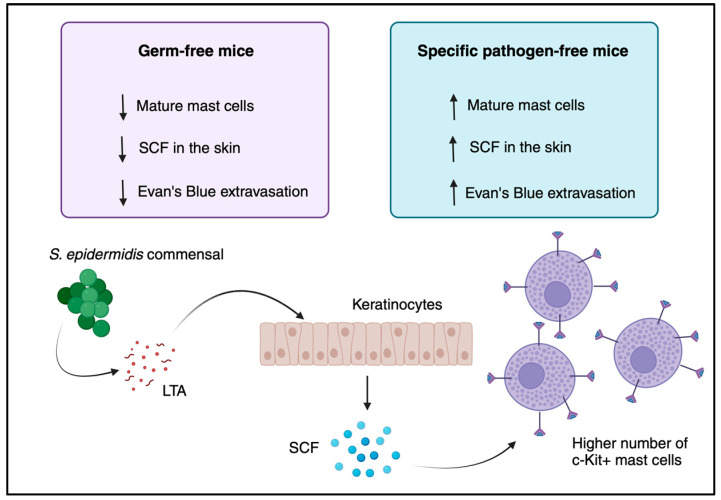
The commensal skin microbiome fosters the maturation of mast cells. In a typical, specific-pathogen-free mouse, the presence of lipoteichoic acid (LTA) from the skin microbiome prompts keratinocytes to produce an ample amount of stem cell factor (SCF), which leads to the correct maturation of mast cells. However, in germ-free mice, mast cells exhibit reduced expression of the c-Kit receptor because there is an insufficient supply of SCF. Figure created with BioRender.

**Figure 4 cells-12-02624-f004:**
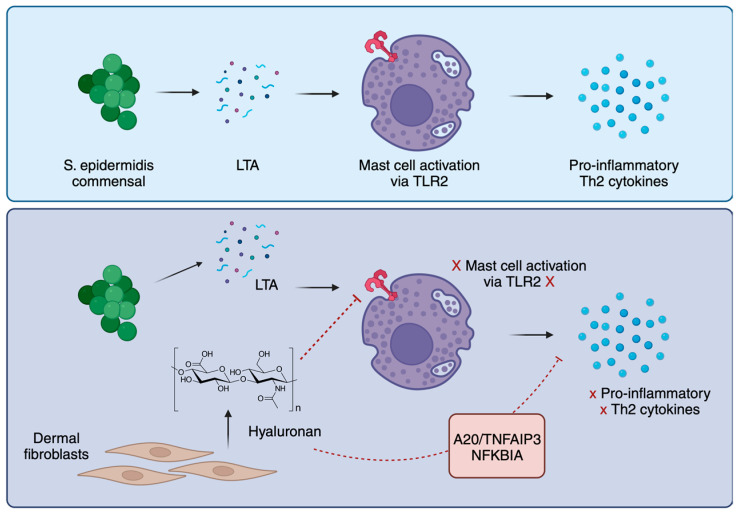
Fibroblasts control mast cell tolerance to commensal bacteria via the production of hyaluronan. Skin mast cells exhibit tolerance to commensal bacteria through interactions with dermal fibroblasts. These fibroblasts effectively regulate mast cell reactivity by increasing the expression of the A20 inhibitor of NF-κB. This intricate mechanism serves to preserve skin homeostasis while permitting mast cell responses to other external challenges. Figure created with BioRender.

**Figure 5 cells-12-02624-f005:**
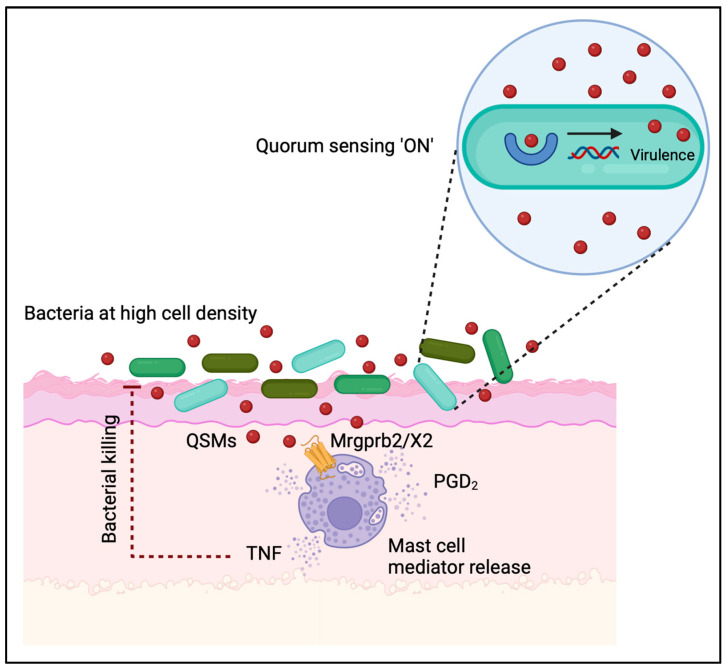
Mast cells are capable of sensing interbacterial communication via the MRGPR receptors. Bacteria produce soluble quorum-sensing molecules (QSMs) to signal their population density. When the bacterial population reaches a critical mass, these QSMs activate specific bacterial genes related to virulence and pathogenicity. Human MRGPRX2 and mouse Mrgprb2, which are receptors specific to mast cells, can recognize and bind cationic bacterial QSMs originating from Gram-positive bacteria. This recognition triggers rapid degranulation of mast cells, leading to the release of various mediators, i.e., antibacterial granular content and prostaglandin D_2_ (PGD_2_). This antibacterial response results in the destruction of bacteria and initiates other immune responses with antibacterial properties. Figure created with BioRender.

**Table 1 cells-12-02624-t001:** Differences between tryptase–chymase- and tryptase-rich mast cells in humans.

Cytokine/Chemokine/Complement Receptors	MC_TC_ (Connective)	MC_T_ (Mucosal)
Tryptase	High	High
Chymase	High	Low
Heparin	Low	High
MrgprX2	High	Negative
External TLR-1/-2/-4/-5/-6	Low	High
Internal TLR-3/-8	High	High
Internal TLR-7/-10	Low	High
Internal TLR-9	High	Negative
C3aR	High	Low
C5aR	High	Low/negative
Histamine receptor-1/-2	High	Low
Histamine receptor-3/-4	Low	High
CMA1, HEY1, and C5R1	High	Negative

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
