# Peer review of "Emerging Role of the Mast Cell–Microbiota Crosstalk in Cutaneous Homeostasis and Immunity"

_cells, 2023, doi:10.3390/cells12222624_

Round 1

Reviewer 1 Report

Comments and Suggestions for Authors

The authors describe the state of knowledge on MC/microbiota crosstalk in the skin. The literature is not abundant and there are many hypotheses on this interaction. The authors provide a coherent overview of the subject.

Some points

-Lines 26 à 28 : A rather misleading list that mixes immune and non-immune cells. Why aren't CD4 and CD8 T lymphocytes mentioned?

-Line 29 the surface of the skin, not the skin itself

-Lines 51-52 Mast cells possess a spatial advantage by strategically positioning themselves near blood vessels and nerves, enabling them to quickly detect and respond to foreign substances above all facilitating communication with blood vessels cells such as endothelial cells and pericyte and sensitive neurons.

-Line 71 leukocyte rather than immunocyte

-Figure 1 needs more detailed caption and correction

I think the figure should be corrected by using ref St John AL, Rathore APS, Ginhoux F. New perspectives on the origins and heterogeneity of mast cells. Nat Rev Immunol. 2023 Jan;23(1):55-68. doi: 10.1038/s41577-022-00731-2. Epub 2022 May 24. PMID: 35610312.

 and P. valent, Theranostics 2020, vol 10 issue 23

To me BMCP stands for basophil–mast cell common progenitor (BMCP)

The notion of MC/moncocyte committed progenitor is not commonly accepted

-Line 213-232 interesting but does not involve MCs

-In Fig 2 D Il-1b drawn as Abs is misleading. MCETosis is very controversial in MCs, is panel D very useful here?

Rather MC activation via SP/MRGPRX2 would be more relevant/ last chapters

-Line 277 are hyperactivated by LPS and not IL-33/LPS

Comments on the Quality of English Language

minor editing

Some sentences are a little awkward, but overall it's clear.

Author Response

We thank you and the reviewers for your valuable time and effort in facilitating the review of our manuscript titled ‘Emerging Role of the Mast Cell-Microbiota Crosstalk in Cutaneous Homeostasis and Immunity’ (cells-2692568). We have addressed all the concerns and have a revised manuscript that encompasses one new figure and additional text on mast cell interaction with melanocytes and potential link between mast cells and commensals in relation to innate memory. Major changes in the manuscript have been highlighted in yellow. The point-by-point responses are included herein.

REVIEWER #1

The authors describe the state of knowledge on MC/microbiota crosstalk in the skin. The literature is not abundant and there are many hypotheses on this interaction. The authors provide a coherent overview of the subject.

Authors’ response: Thank you for taking the time to read our manuscript and offer your criticisms. Your constructive feedback has allowed us to generate a revised manuscript that we feel is of greater quality and clarity.

Reviewer’s comment 1: Lines 26 à 28 : A rather misleading list that mixes immune and non-immune cells. Why aren't CD4 and CD8 T lymphocytes mentioned?

Authors’ response: Thank you for this valuable critique. We have edited the statement, and the changes are reflected in lines 28-29.

Reviewer’s comment 2: Line 29 the surface of the skin, not the skin itself

Authors’ response: Thank you for pointing this out. We have edited line 30 to highlight the change.

Reviewer’s comment 3: Lines 51-52 Mast cells possess a spatial advantage by strategically positioning themselves near blood vessels and nerves, enabling them to quickly detect and respond to foreign substances above all facilitating communication with blood vessels’ cells such as endothelial cells and pericyte and sensitive neurons.

Authors’ response: Thank you for your suggestion. We have edited lines 53-55 to include these changes.

Reviewer’s comment 4: Line 71 leukocyte rather than immunocyte

Authors’ response: Thank you for bringing this to our attention. We have updated the text to address this concern (line 73).

Reviewer’s comment 5: Figure 1 needs more detailed caption and correction

I think the figure should be corrected by using ref St John AL, Rathore APS, Ginhoux F. New perspectives on the origins and heterogeneity of mast cells. Nat Rev Immunol. 2023 Jan;23(1):55-68. doi: 10.1038/s41577-022-00731-2. Epub 2022 May 24. PMID: 35610312.

and P. valent, Theranostics 2020, vol 10 issue 23

To me BMCP stands for basophil–mast cell common progenitor (BMCP)

The notion of MC/moncocyte committed progenitor is not commonly accepted

Authors’ response: We agree with the reviewer that this is an important information, and the Fig.1 has been updated accordingly.

Reviewer’s comment 6: Line 213-232 interesting but does not involve MCs

Authors’ response: Thank you for pointing out this concern and for the opportunity to clarify. Here, we have tried to discuss the various mechanisms by which commensal microflora could exert its effects on the immune system, mechanisms that could be explored in the context of mast cells later.

Reviewer’s comment 7: In Fig 2 D Il-1b drawn as Abs is misleading. MCETosis is very controversial in MCs, is panel D very useful here?  Rather MC activation via SP/MRGPRX2 would be more relevant/ last chapters

Authors’ response: Thank you for your valuable critique. We have taken your thoughtful suggestion and edited Fig. 2 to incorporate these changes.

Reviewer’s comment 8: Line 277 are hyperactivated by LPS and not IL-33/LPS

Authors’ response: Thank you for bringing this to our attention. We have updated the text to address this concern (line 403).

Reviewer’s comment 9: Comments on the Quality of English Language - minor editing. Some sentences are a little awkward, but overall it's clear.

Authors’ response: We have taken this critique into consideration and edited the manuscript for the English syntax.

Reviewer 2 Report

Comments and Suggestions for Authors

Bosveld, C.J., et al. present a review of recent advances in the relationship between mast cells and skin microorganisms in physiological conditions and, also, in some common skin pathologies. The review present interesting and relevant evidences of cross-talk between MC and skin microbiota, which is quite relevant in certain inflammatory skin diseases.

Comments:

1. Lines 77 to 80 make reference to the classic division of MC phenotypes in mice. Molecular differences between those types have been recently found by Tauber, M J.Exp Med 220(10):e20230570, 2023. Data on the expression of MRGPR receptors as markers of distinct subtypes of MCs in mice, together with the evidence indicating distinct subtypes in humans must be mentioned. In the same line, Figure 1 must include the expression of MRGPR receptors as markers of MC subtypes in mice and other possible markers in humans.

2. Lines 130 to 143 make mention of the interaction between MC and other cells in the skin, however, interactions of MC and melanocytes has been proposed as an important factor leading to an increase of melanoma in mastocytosis patients. Interactions between MC and melanocytes must be described with appropriate references.

3. Figure 2 shows mast cell activation by different immunologic stimuli. However, stimulation by neuropeptides, (such as Substance P), secretagoges and ultraviolet light should be included since those stimuli are relevant to activation of MC in the skin. MGPR receptors should be included as main activators of MCs in the skin.

4. Data regarding the role of quorum-sensing molecules (QSM) is quite relevant and has been underscored in the literature. A figure showing the influence of QSM on MC-bacteria communication would be desirable.

5. Discussion on MC-cutaneous microbiome communication could be enriched due to the fact skin MC seem to survive long time and MC present innate immunity memory. Authors must mention this in the discussion or in the concluding remarks and present evidence (if any) of epigenetic modifications induced by microbiota in MC.

Author Response

We thank you the editor and the reviewers for your valuable time and effort in facilitating the review of our manuscript titled ‘Emerging Role of the Mast Cell-Microbiota Crosstalk in Cutaneous Homeostasis and Immunity’ (cells-2692568). We have addressed all the concerns and have a revised manuscript that encompasses one new figure and additional text on mast cell interaction with melanocytes and potential link between mast cells and commensals in relation to innate memory. Major changes in the manuscript have been highlighted in yellow. The point-by-point responses are included herein.

REVIEWER #2

Bosveld, C.J., et al. present a review of recent advances in the relationship between mast cells and skin microorganisms in physiological conditions and, also, in some common skin pathologies. The review present interesting and relevant evidences of cross-talk between MC and skin microbiota, which is quite relevant in certain inflammatory skin diseases.

Authors’ response: We thank the reviewer for their enthusiasm and support for our work. The concerns outlined below have been addressed in the revised manuscript. 

Reviewer’s comment 1: Lines 77 to 80 make reference to the classic division of MC phenotypes in mice. Molecular differences between those types have been recently found by Tauber, M J.Exp Med 220(10):e20230570, 2023. Data on the expression of MRGPR receptors as markers of distinct subtypes of MCs in mice, together with the evidence indicating distinct subtypes in humans must be mentioned. In the same line, Figure 1 must include the expression of MRGPR receptors as markers of MC subtypes in mice and other possible markers in humans.

Authors’ response: Thank you for pointing out this study. We have added data from Tauber et al. to lines 86-91 and Fig 1.

Reviewer’s comment 2: Lines 130 to 143 make mention of the interaction between MC and other cells in the skin, however, interactions of MC and melanocytes has been proposed as an important factor leading to an increase of melanoma in mastocytosis patients. Interactions between MC and melanocytes must be described with appropriate references.

Authors’ response: Thank you for raising this important point. We have added details on mast cell-melanocyte interactions in lines 156-165.

Reviewer’s comment 3: Figure 2 shows mast cell activation by different immunologic stimuli. However, stimulation by neuropeptides, (such as Substance P), secretagoges and ultraviolet light should be included since those stimuli are relevant to activation of MC in the skin. MGPR receptors should be included as main activators of MCs in the skin.

Authors’ response: Agreed, and we have addressed this by editing the figure. The new figure 2 reflects this change.

Reviewer’s comment 4: Data regarding the role of quorum-sensing molecules (QSM) is quite relevant and has been underscored in the literature. A figure showing the influence of QSM on MC-bacteria communication would be desirable.

Authors’ response: Thank you for your excellent suggestion. We have included a new figure (Fig. 5) showing the influence of QSM on MC-bacteria communication in the revised manuscript.

Reviewer’s comment 5: Discussion on MC-cutaneous microbiome communication could be enriched due to the fact skin MC seem to survive long time and MC present innate immunity memory. Authors must mention this in the discussion or in the concluding remarks and present evidence (if any) of epigenetic modifications induced by microbiota in MC.

Authors’ response: Another excellent suggestion. We have added greater detail to the discussion regarding the innate immune memory presented by mast cells and potential effect of microbiota-induced epigenetic modification in mast cell function (lines 512-545).

Again, we thank the reviewers for highlighting any point that is unclear as it has helped the manuscript tremendously.